# A Comparison of Three Measures to Identify Postnatal Anxiety: Analysis of the 2020 National Maternity Survey in England

**DOI:** 10.3390/ijerph19116578

**Published:** 2022-05-28

**Authors:** Gracia Fellmeth, Siân Harrison, Maria A. Quigley, Fiona Alderdice

**Affiliations:** 1NHIR Policy Research Unit in Maternal and Neonatal Health and Care, National Perinatal Epidemiology Unit, Nuffield Department of Population Health, University of Oxford, Oxford OX3 7LF, UK; sian.harrison@npeu.ox.ac.uk (S.H.); maria.quigley@npeu.ox.ac.uk (M.A.Q.); fiona.alderdice@npeu.ox.ac.uk (F.A.); 2School of Nursing and Midwifery, Queen’s University Belfast, Belfast BT7 1NN, UK

**Keywords:** postnatal, anxiety, identifying, screening, GAD, EPDS

## Abstract

Perinatal anxiety affects an estimated 15% of women globally and is associated with poor maternal and infant outcomes. Identifying women with anxiety is essential to prevent these adverse associations, but there are a number of challenges around measurement. We used data from England’s 2020 National Maternity Survey to compare the prevalence of anxiety symptoms at six months postpartum using three different measures: the two-item Generalised Anxiety Disorders Scale (GAD-2), the anxiety subscales of the Edinburgh Postnatal Depression Scale (EPDS-3A) and a direct question. The concordance between each pair of measures was calculated using two-by-two tables. Survey weights were applied to increase the representativeness of the sample and reduce the risk of non-response bias. The prevalence of postnatal anxiety among a total of 4611 women was 15.0% on the GAD-2, 28.8% on the EPDS-3A and 17.1% on the direct question. Concordance between measures ranged between 78.6% (95% CI 77.4–79.8; Kappa 0.40) and 85.2% (95% CI 84.1–86.2; Kappa 0.44). Antenatal anxiety was the strongest predictor of postnatal anxiety across all three measures. Women of Black, Asian or other minority ethnicity were less likely to report self-identified anxiety compared with women of White ethnicity (adjusted odds ratio 0.44; 95% CI 0.30–0.64). Despite some overlap, different anxiety measures identify different groups of women. Certain population characteristics such as women’s ethnicity may determine which type of measure is most likely to identify women experiencing anxiety.

## 1. Introduction

Anxiety disorders are common during the perinatal period [1]. Anxiety disorders are characterised by core symptoms of anxiety, including cognitive distortions, physiological arousal and behavioural avoidance [2]. Globally, an estimated 15% of women experience anxiety symptoms postnatally, with significantly higher rates in low- and middle-income countries (LMIC) [3,4,5]. Figures from the UK are slightly lower, with one review estimating prevalence to be 12–15% during pregnancy and 8% postnatally [6]. Exposure to the physiological and psychosocial impacts of anxiety during this critical period has been associated with poor maternal, infant and child outcomes, including delayed cognitive and behavioural development [7,8,9]. Timely identification and treatment of perinatal anxiety are therefore essential to prevent adverse outcomes for women and their children.

A number of challenges exist around measuring perinatal anxiety [2]. Although several self-report measures have been validated to facilitate the recognition of anxiety, uncertainty remains around which instrument is most suitable for perinatal women and best able to identify women who need support [10]. The number of items included on a measure and the time required for completion have been highlighted as barriers to their administration. Shortened measures have been suggested as a solution to this. In the UK, the National Institute for Health and Care Excellence (NICE) recommends administering the two-item version of the Generalised Anxiety Scale (GAD-2) to all women at their first antenatal appointment and in the early postnatal period [11,12]. However, there remains uncertainty around the psychometric properties of the GAD-2 in perinatal populations, and direct comparisons of how the GAD-2 performs against other self-report measures are lacking.

An alternative method of identifying women with anxiety is to ask women directly whether they self-identify as having anxiety. This measure of ‘self-identified anxiety’ may offer advantages in certain contexts. For instance, it may help to identify women who have anxiety but whose symptoms differ from those captured by self-report measures. Self-identified anxiety may also help to identify women with anxiety who do not meet the severity threshold of standardised measures. A direct question may be preferable in communities less familiar with the culture of ‘test-taking’ [10,13,14]. In a previous study of postnatal women in the UK, almost half (42%) of women who self-identified as having anxiety scored below the threshold on the anxiety subscale of the Edinburgh Postnatal Depression Scale (EPDS-3A) [15]. Results suggested that eliciting women’s own views of their psychological wellbeing through a direct question may help to ensure that more women experiencing symptoms of anxiety are identified and offered appropriate support.

In this analysis, we compared the prevalence of anxiety symptoms identified using the GAD-2—the currently recommended measure for postnatal women in the UK [11]—with the prevalence of anxiety symptoms identified using the EPDS-3A and using a direct question. We use data from England’s 2020 National Maternity Survey (NMS), which captures the experiences of women who gave birth during the first wave of the COVID-19 pandemic in May 2020 [16]. We built upon existing work by assessing symptoms of anxiety at six months postpartum, where previous analyses focused on earlier postnatal periods. By directly comparing the GAD-2 with another standardised measure as well as with a direct question, we built on previous studies that were limited to comparing self-identified anxiety against a single standardised measure. Our aims were to determine the prevalence of anxiety symptoms identified by the GAD-2, EPDS-3A and a direct question; to assess the extent of concordance between the three measures; and to compare the characteristics of women with anxiety on each of the three measures.

## 2. Materials and Methods

### 2.1. Study Setting and Participants

We conducted an analysis of data from the 2020 NMS in England. The survey methods have been described in detail elsewhere [16]. In summary, a random population-based sample of 16,050 women aged 16 years or older who were living in England and had given birth during a two-week period in May 2020 was identified by the Office for National Statistics (ONS) using birth registration records. Women were sent questionnaires six months after they had given birth. The survey included questions on care during pregnancy, labour and birth, and the postnatal period and included mental health outcomes. Women had a choice of completing questionnaires on paper, online or over the telephone with an interpreter if required. Reminder packs were sent to non-respondents using a tailored reminder system [17].

### 2.2. Anxiety Measures

Self-identified anxiety was assessed using a single, direct question asking women whether they had experienced anxiety in the postpartum period, worded as follows: ‘Did you experience any of the following after the birth of your baby?’. Anxiety was listed as one of the conditions. Women were asked to indicate if they had experienced anxiety at one month, three months and/or six months after the baby’s birth. Responses were coded as either ‘yes’ or ‘no’ (binary) for each time point. Because surveys were sent to women at approximately six months postpartum, women’s responses about experiencing anxiety at one and three months postpartum relied on their recall. In order to minimise the risk of recall bias, we used women’s responses about anxiety at six months postpartum as our measure of self-identified anxiety. This also maximised comparability with the two standardised measures, which were also administered at six months postpartum. For the purpose of our analyses, therefore, women who responded ‘yes’ to having anxiety at six months postpartum were defined as having self-identified anxiety.

We used two standardised self-report measures of anxiety: the GAD-2 and the EPDS-3A. The GAD-2 is a shortened version of the original seven-item Generalised Anxiety Disorder scale (GAD) [18]. The GAD is used to identify symptoms of anxiety and is designed for use in the general (non-perinatal) population. The GAD-2 asks respondents to rate the frequency with which they have experienced the following two symptoms of anxiety over the previous two weeks: *feeling nervous, anxious, or on edge* and *not being able to stop or control worrying*. Each item is scored on a four-point Likert scale from 0 (not at all) to 3 (nearly every day), with the total score ranging from 0 to 6. A score of ≥3 on the GAD-2 was identified as an acceptable cut-off for identifying clinically significant anxiety symptoms in the general population, with sensitivity and specificity of 86% and 83%, respectively [19]. In our analysis, GAD-2 scores ≥3 were considered to suggest possible clinically significant anxiety.

The Edinburgh Postnatal Depression Scale (EPDS) is one of the most widely used screening instruments for depression during the perinatal period [20]. The original scale consists of ten items that ask women to rate the intensity of depressive symptoms they have experienced within the previous seven days. Each item is scored 0–3 with a maximum total score of 30 and higher scores representing greater symptom severity. Items 3, 4 and 5 of the EPDS assessed symptoms of anxiety and were administered as a stand-alone anxiety subscale (EPDS-3A) [21,22]. The three items of the EPDS-3A are as follows: *I have blamed myself unnecessarily when things went wrong*; *I have been anxious or worried for no good reason*; and *I have felt scared or panicky for no very good reason*. Each item was scored on a four-point Likert scale from 0 (never or not at all) to 3 (very often or most of the time). Scores range from 0 to 9, with a threshold of ≥6 considered to indicate possible anxiety [23]. In our analysis, EPDS-3A scores ≥6 were considered indicative of possible clinically significant anxiety. EPDS-3A and GAD-2 items, along with their scoring criteria, are summarised in Appendix A. The three measures of anxiety were presented in the following order in the questionnaire: first the self-identified measure, followed by the EPDS-3A and finally the GAD-2.

### 2.3. Sociodemographic, Clinical and Psychological Variables

We selected socio-demographic, clinical and psychological variables, which are known to be associated with perinatal anxiety. Socio-demographic variables were age (under 25 years; 25–34 years; 35 years and over); education (under 16 years; 17–18 years; 19 years and over); ethnicity (White; Black or minority ethnic (BME)); country of birth (UK; outside of UK); index of multiple deprivation (IMD) based on the area of residence (from most (1) to least (5) deprived); whether the pregnancy was planned (yes; no); and women’s reaction to the pregnancy (pleased or happy; no particular feelings or unhappy). The IMD ranks small geographic areas in England by the level of deprivation, assessed according to the following seven domains: income, employment, education and skills, health and disability, crime, barriers to housing and services and the quality of the local living environment [24]. Clinical variables were multiple births (singleton; multiple), the presence of any chronic health conditions complicating pregnancy or pregnancy-related problems (yes; no) and whether children of participating women required admission to the neonatal intensive care unit (NICU) (yes; no). The psychological variable was antenatal anxiety, assessed using a single direct question as follows: ‘Did you have any mental health problems during your pregnancy?’. Women who selected ‘Yes—anxiety’ were classified as having had antenatal anxiety.

### 2.4. Statistical Analysis

Descriptive characteristics of participants were summarised. Because respondents differed from non-respondents on key socio-demographic characteristics, we calculated non-response weights to adjust the sample in order to increase representativeness and reduce the risk of non-response bias [16]. The weighted prevalence of anxiety symptoms according to each measure was determined along with the proportion of women who reported symptoms on more than one measure. Weighted mean EPDS-3A and GAD-2 scores were calculated for women with and without self-identified anxiety to assess differences in scores between these two groups. The distribution of GAD-2 and EPDS-3A scores and the percentage who scored 0, 1 and 2 on individual items of the scales were plotted. Cronbach’s alpha was calculated for the GAD-2 and EPDS-3A to assess their reliability. Agreement between the measures was assessed by calculating the proportion and 95% confidence intervals (CI) of women with concordance on each pair of measures by summing the diagonal in a two-by-two table. Cohen’s kappa coefficient was used to quantify the statistical agreement between each pair of measures, taking into account the possibility of the agreement occurring by chance [25]. Kappa coefficients were interpreted using the following cut-offs: 0.00–0.20 ‘no agreement’, 0.21–0.39 ‘minimal agreement’, 0.40–0.59 ‘weak agreement’, 0.60–0.79 ‘moderate agreement’, 0.80–0.90 ‘strong agreement’ and >0.90 ‘almost perfect agreement’ [25].

Associations between socio-demographic (age, education, ethnicity, country of birth, IMD, planned pregnancy, reaction to pregnancy), clinical (multiple births, health condition, NICU admission) and psychological (antenatal anxiety) factors and postnatal anxiety were explored in univariable and multivariable logistic regression analyses. Unadjusted and adjusted odds ratios (OR) of associations between these variables and postnatal anxiety were calculated for each of the three anxiety measures. Variables that remained statistically significant at the *p* < 0.05 level in the final adjusted model were considered to be significantly associated with anxiety. Full case analysis was used throughout. Analyses were conducted using STATA version 15 (StataCorp LLC, College Station, TX, USA). All means, proportions and odds ratios were survey-weighted using the *svy* command in STATA.

## 3. Results

Completed questionnaires were returned by 4611 women, giving a valid response rate of 28.9% when excluding those returned as undeliverable [16]. Baseline characteristics of the sample are reported in detail elsewhere [16]. After applying survey weights, a third of women who participated in the survey were aged 30–34 years (33.7%), 69.1% were born in the UK, just under half (47.5%) were living in areas in the two most deprived quintiles on the IMD and 44.3% were primiparous [16]. Anxiety measures were complete for 4508 (97.8%) women; subsequent analyses are based on these complete cases.

### 3.1. Prevalence of Anxiety Symptoms

The weighted prevalence of anxiety symptoms on each measure is summarised in Table 1. At six months postpartum, 17.1% of women had self-identified anxiety, 15.0% had elevated GAD-2 scores and 28.8% had elevated EPDS-3A scores. One-third (36.0%) of women reported anxiety symptoms on at least one measure, and 7.3% reported symptoms on all three measures. Weighted mean GAD-2 scores were 3.04 and 0.83 among women with and without self-identified anxiety, respectively. Weighted mean EPDS-3A scores were 6.27 and 3.49 among women with and without persistent self-identified anxiety, respectively. Among those with self-identified anxiety, these mean scores fall above the cut-off for the GAD-2 and EPDS-3A. Figure 1 shows the distribution of total GAD-2 and EPDS-3A scores, and Figure 2 shows the distribution of responses to individual question items on the GAD-2 and EPDS-3A. Lower total scores were seen on the GAD-2, with the majority of women scoring zero and few scoring above the threshold of 3. Higher total scores were seen on the EPDS-3A, with fewer women scoring zero. When comparing the individual items, EPDS item 5 (*I feel scared or panicky for no good reason*) showed a similar response pattern to GAD-2 items 1 and 2, with more women scoring 0 on this item. Cronbach alpha was 0.88 for the GAD-2 and 0.76 for the EPDS-3A.

### 3.2. Agreement between Anxiety Measures

Table 2, Table 3 and Table 4 show the concordance between each pair of anxiety measures. The concordance between self-identified anxiety and the GAD-2 was 85.2% (95% CI 84.1–86.2; Kappa 0.439). Concordance between self-identified anxiety and the EPDS-3A was 78.6% (95% CI 77.4–79.8; Kappa 0.399). Concordance between the GAD-2 and EPDS-3A was 79.6% (95% CI 78.4–80.1; Kappa 0.415). The Kappa coefficient for all three measures was 0.414. The Kappa coefficients suggest a ‘weak level of agreement’ between measures. Figure 3 shows the overlap between the three measures.

### 3.3. Characteristics of Women with Anxiety

Table 5 summarises the characteristics of women with anxiety. After controlling for all other variables in the multivariable model, antenatal anxiety was the strongest predictor of postnatal anxiety across all three measures. Antenatal anxiety was associated with an increased likelihood of self-identified anxiety (aOR 5.35; 95% CI 4.37–6.55), elevated EPDS-3A scores (aOR 3.64; 95% CI 3.03–4.36) and elevated GAD-2 scores (aOR 4.16; 95% CI 3.35–5.15). Women of Black, Asian or other minority ethnicity were less likely to report self-identified anxiety compared with women of White ethnicity (aOR 0.44; 95% CI 0.30–0.64), while women born outside of the UK were less likely than women born in the UK to have elevated GAD-2 scores (aOR 0.66; 95% CI 0.49–0.89). Women aged over 35 years were less likely than those aged 25–34 years to have elevated EPDS-3A scores (aOR 0.80; 95% CI 0.67–0.96). Women who were unhappy with or had mixed feelings about their pregnancy were more likely to have elevated GAD-2 scores compared with women who felt pleased about pregnancy (aOR 1.71; 95% CI 1.30–2.25) compared with those who were pleased about their pregnancy. Women with a health condition were more likely than those without to have elevated EPDS-3A scores (aOR 1.49; 95% CI 1.16–1.93). 

## 4. Discussion

We compared the prevalence of anxiety symptoms at six months postpartum on two standardised self-report measures and one direct question on self-identified anxiety. We found wide variation in the prevalence depending on the measure used. Prevalence was highest using the EPDS-3A, which yielded an estimate of 28.7%, while prevalence using the GAD-2 was almost half of this at 15.0%. The prevalence of self-identified anxiety was at the lower end of this range, with 17.1% of women reporting anxiety on the direct question. Previous estimates of postnatal anxiety from meta-analyses have reported a prevalence of 15% in high-income settings [3]. These pooled estimates pre-date the COVID-19 pandemic. Our data were collected during the first wave of the COVID-19 pandemic in the UK when higher levels of anxiety might be expected. It is possible that the higher rates seen on the EPDS-3A may reflect this trend, although there is no evidence to suggest that only EPDS-3A scores and not GAD-2 scores or self-identified anxiety would be affected. Alternatively, the high prevalence of symptoms accoding to the EPDS-3A may suggest that this measure is overly inclusive when used with the recommended threshold of ≥6. In the absence of a clinical interview—the ‘gold standard’ for the diagnosis of mental disorders—we cannot conclude whether this is the case or whether, in fact, the GAD-2 is over-excluding women with anxiety. Both overly inclusive and overly exclusive measures are problematic: while the former can result in women being incorrectly identified as having anxiety, creating unnecessary strains on mental health services, the latter risks women with anxiety being missed and left unsupported.

The wording and scoring of EPDS-3A and GAD-2 items may have contributed to the difference in prevalence observed. On both measures, a score of zero denotes ‘never’ or ‘not at all’ experiencing that particular symptom. However, a score of one represents significantly different levels of symptoms on each measure. A score of one on the EPDS-3A is defined as experiencing the symptom ‘not very often’ or ‘hardly ever’, while a score of one on the GAD-2 is defined as experiencing the symptom on ‘several days’ (Appendix A). This may pose a difficulty for women with occasional symptoms of anxiety: these women can select the category of ‘not very often’ or ‘hardly ever’ on the EPDS (a score of one), but on the GAD, they must select between either ‘no symptoms’ (a score of zero) or having symptoms on ‘several days’ (a score of one). It is possible that this larger conceptual gap between a score of zero and one on the GAD-2 may be pushing women with mild anxiety towards selecting zero, thereby underestimating the true prevalence of symptoms.

Although there was overlap between the three measures, the Kappa values were relatively low, corresponding to a weak level of agreement between them. The GAD-2, EPDS-3A and direct questions on anxiety each identified different groups of women. Although the prevalence estimates were similar for the GAD-2 and self-identified anxiety, the overlap between these measures shows that they are not identifying all the same women. Perhaps of greatest concern are the women with self-identified anxiety who are not being identified by either of the standardised measures. Women who report anxiety on a direct question are likely to benefit from support, even without scoring above the EPDS-3A or GAD-2 thresholds. Hence it may be appropriate for women to be asked about self-identify anxiety alongside completing standardised measures in order to avoid missing those who may need follow-up.

When we compared the characteristics of women with anxiety on each of the three measures, the factor most strongly associated with postnatal anxiety across all measures was antenatal anxiety. Antenatal anxiety was associated with an approximately four-fold increase in the likelihood of experiencing postpartum anxiety. The strongest association was between antenatal anxiety and self-identified postnatal anxiety. In part, this might be explained by the fact that antenatal anxiety was also self-identified: women who self-identified as having postnatal anxiety may be most likely to also self-identify as having had antenatal anxiety. The association between antenatal and postnatal anxiety also has important clinical implications, as it suggests a trend of anxiety symptoms that persist throughout the perinatal period. Ideally, women with anxiety should be identified during pregnancy and offered timely support to address symptoms [26]. Routinely asking all women about anxiety symptoms during antenatal appointments, as recommended by NICE, can help to ensure that women with anxiety are identified and supported from an early stage and prevent symptoms from continuing into the postnatal period.

Women from ethnic minority backgrounds were less likely than women of White ethnicity to report self-identified anxiety, while women born outside the UK were less likely to have elevated EPDS-3A and GAD-2 scores compared to women born in the UK. These results suggest that different groups of women may have different preferences for the type of measures used. Our findings are of particular importance given that women from minority groups—including migrant populations and those from low- and middle-income countries of origin—are at greater risk of perinatal mental disorders [5,27,28,29]. Women from minority ethnic backgrounds may be less likely to respond to a direct question on anxiety due to cultural sensitivities, social desirability or stigmatising attitudes around mental disorders [30]. Among some groups, there may also be a lower awareness of what constitutes anxiety, resulting in women who experience symptoms of anxiety not ascribing their symptoms to anxiety. Standardised self-report measures offer an alternative means of bringing to light problematic symptoms without needing to label these as ‘anxiety’. Conversely, standardised measures may fail to identify culturally diverse manifestations of anxiety, which could be contributing to the lower likelihood of women from minority ethnic groups having anxiety on the GAD-2 and EPDS-3A [30]. Future research should examine in more depth the acceptability of different screening measures among populations of diverse ethnicities [26].

Age was also significantly associated with anxiety. Compared with women aged 25–34 years, those aged over 34 years were less likely to have anxiety on the EPDS-3A. Although results for self-identified anxiety according to age did not reach statistical significance, the trend suggests that younger women may feel more at ease in discussing anxiety. Finally, the method of administration may play a role: while some women may prefer the less personal means of disclosure offered by a self-report measure that they complete independently, others may welcome the opportunity for a more personal discussion as offered by a direct question from a health professional [31]. Importantly, a direct question can be nuanced to assess anxiety symptoms over a longer time period and therefore provide a marker of chronicity and severity, while standardised measures provide only a snapshot in time [2]. The impact of mode of administration upon disclosure of anxiety symptoms warrants further research.

## 5. Strengths and Limitations

Our data stem from a large, population-based survey of women across England. To our knowledge, this is the first comparison of two standardised measures—including one that is recommended for routine use in the perinatal period—with a direct question on self-identified anxiety. There are also a number of limitations. One of the main limitations is the low survey response rate of 28.9%. Response rates to the NMS and to surveys generally have been declining over recent decades [32]. There are many possible reasons for this, including increasing demands on people’s time, survey fatigue and concerns around access to and use of personal information [33]. We used evidence-based recommendations to optimise response rates, including offering incentives and sending reminders [34]. Women who were younger, multiparous, not married at the time of registering the birth of their baby and those born outside of the UK were under-represented in the survey. In order to address the under-representation of these groups, survey weights were applied to analyses of prevalence to reduce the effect of non-response bias. The fact that self-identified anxiety was based on a single question while the GAD-2 and EPDS-3A were based on two and three questions, respectively, may have introduced bias and made the self-identified measure less sensitive. Furthermore, the order in which anxiety measures were presented may have introduced a bias, with women possibly being more inclined to report symptoms on the later questions, having been ‘primed’ to think about their mental health. The absence of a diagnostic clinical interview meant we were unable to conclude which of the three measures most accurately identifies women with anxiety. Finally, our self-reported measure of anxiety did not assess the level of impairment resulting from anxiety symptoms. Eliciting the level of distress and impairment associated with symptoms could provide an additional indicator of when further psychological intervention might be called for, and future research would benefit from including such assessments.

## 6. Conclusions

A comparison of three measures of postnatal anxiety suggests that, despite some overlap, different measures identify different groups of women. Certain population characteristics such as women’s ethnicity and age may determine which type of measure is most likely to identify women experiencing anxiety. Our findings suggest that using a direct question alongside a self-report measure such as the GAD-2 may improve the identification of women who need support and highlight the importance of being attentive to what women say rather than relying solely on standardised measures.

## Figures and Tables

**Figure 1 ijerph-19-06578-f001:**
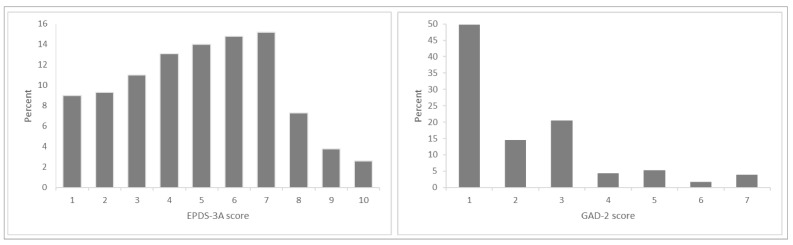
Distribution of total GAD-2 and EPDS-3A scores.

**Figure 2 ijerph-19-06578-f002:**
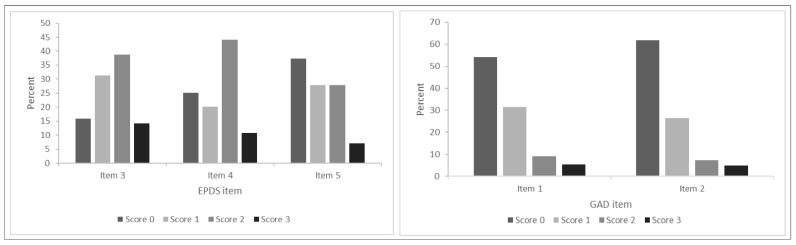
Distribution of scores on individual items of the GAD-2 and EPDS-3A.

**Figure 3 ijerph-19-06578-f003:**
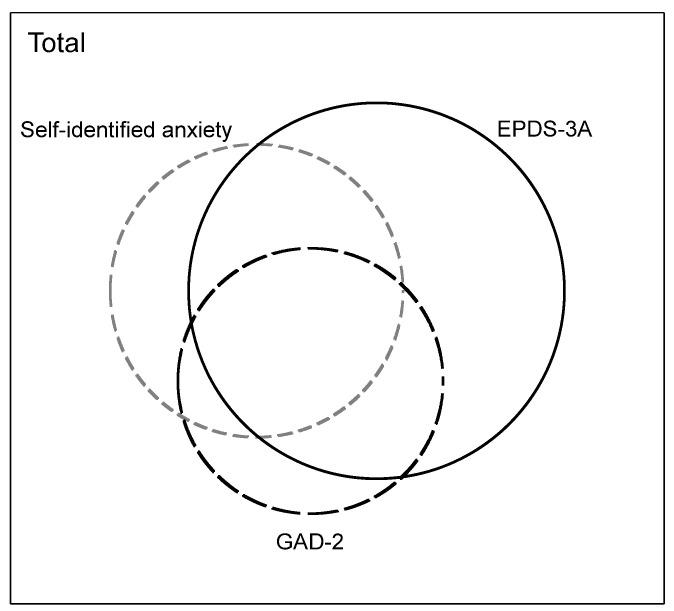
Venn diagram illustrating proportional overlap between self-identified anxiety, GAD-2 and EPDS-3A.

**Table 1 ijerph-19-06578-t001:** Prevalence of postnatal anxiety symptoms on different measures (n = 4508).

Measure	n	%
**Single measure**		
Self-identified anxiety	786	17.1
GAD-2 score ≥3	645	15.0
EPDS-3A score ≥6	1295	28.8
Anxiety on at least one measure	1609	36.0
**Multiple measures**	382	8.5
Self-identified anxiety and GAD-2 score ≥3	382	8.5
Self-identified anxiety and EPDS-3A score ≥6	558	12.1
GAD-2 score ≥3 and EPDS-3A score ≥6	511	11.6
Anxiety on all three measures	334	7.3
No anxiety on any measure	2971	65.9

Note: Prevalence estimates (%) are weighted; counts (n) are unweighted.

**Table 2 ijerph-19-06578-t002:** Concordance and kappa values for self-identified anxiety and GAD-2 (n = 4508).

		GAD-2		Concordance % (95% CI) [Kappa]
		Anxiety	No Anxiety	
**Self-** **identified**	Anxiety	8.5% (382)	8.6% (404)	85.2% (84.1–86.2) [0.439]
	No anxiety	6.5% (263)	76.4% (3459)	

Note: Proportions and Kappa values are weighted; counts are unweighted.

**Table 3 ijerph-19-06578-t003:** Concordance and kappa values for self-identified anxiety and EPDS-3A (n = 4508).

		EPDS-3A		Concordance % (95% CI) [Kappa]
		Anxiety	No Anxiety	
**Self-** **identified**	Anxiety	12.1% (558)	5.0% (228)	78.6% (77.4–79.8) [0.399]
	No anxiety	16.7% (737)	66.2% (2985)	

Note: Proportions and Kappa values are weighted; counts are unweighted.

**Table 4 ijerph-19-06578-t004:** Concordance and kappa values for GAD-2 and EPDS-3A (n = 4508).

		EPDS-3A		Concordance % (95% CI) [Kappa]
		Anxiety	No Anxiety	
**GAD-2**	Anxiety	11.6% (511)	3.4% (134)	79.6% (78.4–80.1) [0.415]
	No anxiety	17.2% (784)	67.8% (3079)	

Note: Proportions and Kappa values are weighted; counts are unweighted.

**Table 5 ijerph-19-06578-t005:** Unadjusted and adjusted odds ratios of associations between participant characteristics and anxiety according to different measures.

	Self-Identified Anxiety (6 m)			EPDS-3A			GAD-2		
	*n* (%)	uOR (95% CI)	aOR (95% CI)	*n* (%)	uOR (95% CI)	aOR (95% CI)	*n* (%)	uOR (95% CI)	aOR (95% CI)
**Age**									
Under 25 years	78 (23.5)	**1.52 (1.12–2.07)**	1.29 (0.91–1.84)	123 (39.4)	**1.63 (1.24–2.14)**	1.32 (0.96–1.82)	75 (22.5)	**1.71 (1.25–2.34)**	1.23 (0.85–1.79)
25–34 years	478 (16.8)	Ref	Ref	800 (28.6)	Ref	Ref	380 (14.5)	Ref	Ref
35 years and over	226 (15.1)	0.89 (0.73–1.07)	0.89 (0.72–1.09)	363 (24.9)	**0.83 (0.70–0.98)**	**0.80 (0.67–0.96)**	183 (12.7)	0.85 (0.69–1.05)	0.88 (0.70–1.10)
**Education**									
Under 16 years	94 (19.4)	1.26 (0.95–1.67)	0.83 (0.61–1.15)	164 (33.2)	**1.41 (1.11–1.79)**	1.05 (0.80–1.37)	94 (19.1)	**1.57 (1.18–2.09)**	1.06 (0.76–0.47)
17–18 years	213 (18.1)	1.16 (0.94–1.42)	0.86 (0.68–1.09)	370 (32.2)	**1.34 (1.13–1.59)**	1.07 (0.88–1.30)	188 (16.7)	**1.33 (1.07–1.66)**	0.99 (0.76–1.28)
19 years and over	473 (16.1)	Ref	Ref	748 (26.2)	Ref	Ref	354 (13.1)	Ref	Ref
**Ethnicity**									
White	726 (19.3)	Ref	Ref	1130 (30.1)	Ref	Ref	577 (16.0)	Ref	Ref
Black or minority ethnic	54 (8.0)	**0.36 (0.26–0.51)**	**0.44 (0.30–0.64)**	148 (23.4)	**0.71 (0.56–0.90)**	0.93 (0.71–1.21)	60 (10.8)	**0.64 (0.46–0.88)**	0.86 (0.59–1.25)
**Country of birth**									
UK	681 (20.0)	Ref	Ref	1097 (31.7)	Ref	Ref	557 (17.4)	Ref	Ref
Outside of UK	102 (10.7)	**0.48 (0.37–0.62)**	0.79 (0.59–1.04)	191 (22.5)	**0.62 (0.51–0.76**)	0.81 (0.64–1.01)	83 (9.6)	**0.50 (0.38–0.66)**	**0.66 (0.49–0.89)**
**IMD**									
1 (most deprived)	120 (17.2)	1.05 (0.79–1.40)	1.06 (0.77–1.46)	209 (30.9)	1.14 (0.90–1.45)	1.05 (0.81–1.36)	123 (18.2)	**1.45 (1.08–1.95)**	1.34 (0.97–1.87)
2	152 (17.4)	1.06 (0.82–1.38)	1.20 (0.91–1.60)	239 (28.7)	1.03 (0.82–1.29)	1.00 (0.78–1.29)	131 (15.6)	1.21 (0.91–1.60)	1.22 (0.90–1.66)
3	166 (17.2)	1.05 (0.81–1.36)	1.04 (0.79–1.36)	266 (27.6)	0.98 (0.79–1.21)	0.94 (0.75–1.18)	137 (14.1)	1.08 (0.82–1.43)	1.05 (0.78–1.41)
4	184 (17.0)	1.04 (0.81–1.33)	1.04 (0.79–1.35)	301 (28.2)	1.01 (0.82–1.23)	1.00 (0.80–1.25)	125 (12.4)	0.93 (0.70–1.23)	0.91 (0.68–1.23)
5 (least deprived)	164 (16.5)	Ref	Ref	280 (28.1)	Ref	Ref	129 (13.2)	Ref	Ref
**Planned pregnancy**									
Planned	608 (16.1)	Ref	Ref	987 (26.7)	Ref	Ref	471 (13.2)	Ref	Ref
Unplanned	175 (20.4)	**1.33 (1.08–1.66)**	1.05 (0.79–1.40)	299 (35.1)	**1.49 (1.24–1.79)**	1.12 (0.89–1.41)	168 (20.1)	**1.65 (1.32–2.06)**	1.01 (0.76–1.36)
**Reaction to pregnancy**									
Pleased or happy	610 (16.0)	Ref	Ref	993 (26.9)	Ref	Ref	461 (12.7)	Ref	Ref
Mixed or unhappy	164 (22.0)	**1.48 (1.18–1.85)**	1.13 (0.85–1.50)	275 (36.3)	**1.55 (1.28–1.88)**	1.22 (0.96–1.55)	173 (23.6)	**2.12 (1.69–2.65)**	**1.71 (1.30–2.25)**
**Multiple birth**									
Singleton	780 (17.3)	Ref	Ref	1273 (28.8)	Ref	Ref	638 (15.1)	Ref	Ref
Twin	6 (9.3)	0.49 (0.18–1.35)	0.56 (0.18–1.78)	15 (26.2)	0.88 (0.41–1.85)	0.89 (0.39–2.01)	5 (9.1)	0.56 (0.19–1.68)	0.40 (0.10–1.62)
**Antenatal anxiety**									
No	386 (10.4)	Ref	Ref	770 (22.0)	Ref	Ref	316 (9.5)	Ref	Ref
Yes	398 (40.9)	**5.93 (4.89–7.18)**	**5.35 (4.37–6.55)**	517 (53.2)	**4.03 (3.39–4.80)**	**3.64 (3.03–4.36)**	327 (34.5)	**5.00 (4.08–6.13)**	**4.16 (3.35–5.15)**
**Chronic health conditions**									
No	677 (16.3)	Ref	Ref	1097 (27.2)	Ref	Ref	544 (14.1)	Ref	Ref
Yes	107 (23.3)	**1.56 (1.19–2.05)**	1.16 (0.85–1.59)	189 (41.0)	**1.87 (1.48–2.35)**	**1.49 (1.16–1.93)**	99 (22.1)	**1.73 (1.31–2.27)**	1.21 (0.89–1.65)
**NICU admission**									
No	688 (18.7)	Ref	Ref	1148 (28.7)	Ref	Ref	577 (15.2)	Ref	Ref
Yes	95 (16.9)	1.13 (0.85–1.51)	1.01 (0.74–1.39)	141 (29.6)	1.05 (0.81–1.36)	0.94 (0.71–1.26)	67 (15.0)	1.01 (0.72–1.41)	0.86 (0.61–1.20)

**Notes:** Bold denotes statistical significance at *p* < 0.05 level. **Abbreviations:** aOR adjusted odds ratio; CI confidence interval; EPDS-3A Edinburgh Postnatal Depression Scale anxiety subscale; GAD-2 two-item Generalised Anxiety Disorder scale; IMD index of multiple deprivation; m months; NICU neonatal intensive care unit; Ref reference category; uOR unadjusted odds rat.

## Data Availability

Data are archived by the NPEU at the University of Oxford. Requests for any data access can be made to the Director of the NPEU. Any requests will be considered by the NPEU data access committee following the NPEU data sharing policy and will be subject to further regulatory approval should access be required for any purposes other than those outlined in the NMS study protocol.

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
