# Peer review of "A Comparison of Three Measures to Identify Postnatal Anxiety: Analysis of the 2020 National Maternity Survey in England"

_ijerph, 2022, doi:10.3390/ijerph19116578_

Round 1

Reviewer 1 Report

The authors presented a well-written original article regarding, A comparison of three measures to identify postnatal anxiety:  Analysis of the 2020 National Maternity Survey in England.

An interesting premise for comparing three measures: two measures were validated questionnaires, and the other one was a "self-reported measure." 

There are possibilities of biases as the self-reported measure has a binary measurement.

Please explain further why the low response rate of 28.9% in this study?

Is it possible to include the prevalence of postnatal anxiety in England and compare it to current global data? 

Please kindly add the definition of postnatal anxiety to the writing.

For references, please kindly recheck and follow the journal standards.

Thank you.

Author Response

Dear Reviewer,

Many thanks for your helpful comments on our manuscript. We have now addressed the issues raised, as follows:

1. There are possibilities of bias as the self-reported measure has a binary measurement.

Thanks for this. All three measures are analysed as binary variables. The self-report measure is based on a single question (with response options yes/no), while the GAD-2 and EPDS-3A are based on two and three questions, respectively, and then converted into binary variables using a pre-defined cut-off score. We agree that the self-report measure may be a less sensitive measure as a result of being based on only one question, and we have added this point to our Limitations section:

“The fact that the self-report measure is based on a single question, while the GAD-2 and EPDS-3A are based on two and three questions, respectively, may have introduced bias and rendered the self-report measure less sensitive.”

2. Please explain further why the low response rate of 28.9% in this study?

We have added the following statement to our Limitations section, to address and provide some possible explanations for the low response rate:

“Response rates to the NMS and to surveys generally have been declining over recent decades. There are many possible reasons for this including increasing demands on people's time, survey fatigue and concerns around access to and use of personal information.”

3. Is it possible to include the prevalence of postnatal anxiety in England and compare it to current global data?

Thank you for this comment. We have now added the prevalence of postnatal anxiety in England to the Introduction: 

“Figures from the UK are slightly lower, with one review estimating prevalence to be 12-15% during pregnancy and 8% postnatally.”

4. Please kindly add the definition of postnatal anxiety to the writing.

We have now added a definition in the first paragraph of the Introduction, as follows:     

“Anxiety disorders are characterised by core symptoms of anxiety including cognitive distortions, physiological arousal and behavioural avoidance.”

5. For references, please kindly recheck and follow the journal standards.

Thanks for flagging this – we have reformatted these.

Reviewer 2 Report

The study aimed to determine the prevalence of anxiety symptoms identified by the GAD-2, EPDS-3A and a direct question as well as assess the concordance between the three measures.

Presented study, especially in case of obtained results (during COVID 19 pandemic first wave) and discussion of the scientific problem, presents high value. The manuscript needs only a minor revision in the methods and results sections.

Materials and Methods

Sociodemographic, clinical and psychological variables

- Better explanation of the index of multiple deprivation IMD should be provided because this index is not used outside of England. The authors do not show key elements of that index and only present the appropriate scale.

- Authors presented clinical variables in which admission to the neonatal intensive care unit was taken into account. It needs to be better explained in case of who was admitted into NICU – women when she was a child (after birth) or child of that adult woman after birth, in which case the woman is also admitted into NICU.

Results

Prevalence of anxiety symptoms

- The authors provided the statement about the weighted mean of GAD-2 and EPDS-3A, but it is unclear what the presented values represent; moreover, these values do not relate directly to any of the figures and tables in the manuscript.

Author Response

Dear Reviewer 2,

Many thanks for your helpful comments on our manuscript. We have now addressed the points raised, as follows:

1. Sociodemographic, clinical and psychological variables: Better explanation of the index of multiple deprivation IMD should be provided because this index is not used outside of England. The authors do not show key elements of that index and only present the appropriate scale.

Many thanks for highlighting this omission. We have now added an explanation of the IMD in the Methods section, as follows:

“The IMD ranks small geographic areas in England by the level of deprivation, assessed according to the following seven domains: income; employment; education and skills; health and disability; crime; barriers to housing and services; and the quality of the local living environment.”

2. Authors presented clinical variables in which admission to NICU was taken into account. It needs to be better explained in case of who was admitted into NICU – women when she was a child (after birth) or child of that adult woman?

This variable refers to infants of women who participated in the survey being admitted to NICU. We have now clarified this in the Methods by amending the sentence as follows:

“…whether children of participating women required admission to the neonatal intensive care unit (NICU) (yes; no).”

3. Results: Prevalence of anxiety symptoms. The authors provided the statement about the weighted mean of GAD-2 and EPDS-3A, but it is unclear what the presented values represent. Moreover, these values do not relate directly to any of the figures and tables in the manuscript.

We have now clarified in the Statistical Analysis section that we present weighted prevalence and means in order to adjust for non-response bias, as follows:

“Because respondents differed to non-respondents on key sociodemographic characteristics, we calculated non-response weights to adjust the sample in order to increase representativeness and reduce the risk of non-response bias. The weighted prevalence of anxiety symptoms according to each measure was determined along with the proportion of women who reported symptoms on more than one measure. Weighted mean EPDS-3A and GAD-2 scores were calculated for women with and without self-identified anxiety to assess differences in scores between these two groups.”

The mean scores did not fit neatly into any of the existing tables, and as they are only two means we did not feel they warranted their own table. Therefore, we report these results only in the text. 

Reviewer 3 Report

Dear editor,

I have found this manuscript is well-written.  I have some comments as shown below.

  1. In administering the anxiety measurement, please describe in more detail as to how the authors manage it. Did they put in order? Which was the first one? Also, please comment on the possible response bias based on the procedure the authors had conducted.
  2. Cronbach’s alpha of both measurements should be provided.
  3. I agree that using the self- identified anxiety may be useful, but it should not report just only symptom, which the authors can obtain from those two measurements. If new items should be added alongside, they would be items addressing on feeling distress due to such anxiety symptoms or on functional impairment, that would urge clinicians to seek further psychological intervention.

Author Response

Dear Reviewer 3, 

Many thanks for reviewing our manuscript. We have address each of the issues raised, as follows:

1. In administering the anxiety measurement, please describe in more detail as to how the authors manage it. Did they it in order? Which was the first one? Also, please comment on the possible response bias based on the procedure the authors has conducted.

Thanks for raising this. In the Methods section, we have now reported the order as follows:

“The three measures of anxiety were presented in the following order in the questionnaire: self-identified, EPDS-3A, GAD-2.”

We agree that the order may have introduced bias in responses and have addressed this in our Limitations sections:

“Furthermore, the order in which the anxiety measures were presented may have introduced a bias, with women possibly being more inclined to report symptoms on the later questions, having been ‘primed’ to think about the mental health.”

2. Cronbach’s alpha of both measurements should be provided.

Many thanks. We have now added the Cronbach’s alpha values for the GAD-2 and the EPDS-3A in the results section.

3. I agree that using the self-identified anxiety may be useful, but it should not report just only symptom, which the authors can obtain from those two measurements. If new items should be added alongside, they would be items addressing on feeling distress due to anxiety or on functional impairment, that would urge clinicians to seek further psychological intervention.

Thanks for this helpful suggestion. We agree this would be a valuable additional issue to ask about. We have listed this in our Limitations section as something that would have been useful to explore, and that future research should focus on:

“Finally, our self-reported measure of anxiety did not assess the level of impairment resulting from anxiety symptoms. Eliciting the level of distress and impairment associated with symptoms could provide an additional indicator of when further psychological intervention might be called for, and future research would benefit from including such assessments.”